# Combination, Modulation and Interplay of Modern Radiotherapy with the Tumor Microenvironment and Targeted Therapies in Pancreatic Cancer: Which Candidates to Boost Radiotherapy?

**DOI:** 10.3390/cancers15030768

**Published:** 2023-01-26

**Authors:** Sofian Benkhaled, Cedric Peters, Nicolas Jullian, Tatjana Arsenijevic, Julie Navez, Dirk Van Gestel, Luigi Moretti, Jean-Luc Van Laethem, Christelle Bouchart

**Affiliations:** 1Department of Radiation Oncology, Hopital Universitaire de Bruxelles (H.U.B.), Institut Jules Bordet, Université Libre de Bruxelles (ULB), Rue Meylenmeersch 90, 1070 Brussels, Belgium; 2Department of Radiation Oncology, UNIL-CHUV, Rue du Bugnon 46, 1011 Lausanne, Switzerland; 3Department of Radiation Oncology, AZ Turnhout, Rubensstraat 166, 2300 Turnhout, Belgium; 4Laboratory of Experimental Gastroenterology, Université Libre de Bruxelles (ULB), 1070 Brussels, Belgium; 5Department of Gastroenterology, Hepatology and Digestive Oncology, Hopital Universitaire de Bruxelles H.U.B. CUB Hopital Erasme, Université Libre de Bruxelles (ULB), Route de Lennik 808, 1070 Brussels, Belgium; 6Department of Hepato-Biliary-Pancreatic Surgery, Hopital Universitaire de Bruxelles H.U.B. CUB Hopital Erasme, Université Libre de Bruxelles (ULB), 1070 Brussels, Belgium

**Keywords:** pancreatic cancer, radiotherapy, immunotherapy, immune checkpoint inhibition, tumor microenvironment, targeted therapy

## Abstract

**Simple Summary:**

Progress in pancreatic ductal adenocarcinoma cancer (PDAC) has been, and still is, difficult. Compared to other solid tumors, the PDAC’s tumor microenvironment (TME) is unique and complex and prevents systemic agents from effectively penetrating and killing tumor cells. Radiotherapy (RT) has the potential to modulate the TME and therefore enhance the effectiveness of targeted systemic therapies. The success of future clinical trials will depend on our understanding of the complex and dynamic TME and therapy associations, including RT.

**Abstract:**

Pancreatic ductal adenocarcinoma cancer (PDAC) is a highly diverse disease with low tumor immunogenicity. PDAC is also one of the deadliest solid tumor and will remain a common cause of cancer death in the future. Treatment options are limited, and tumors frequently develop resistance to current treatment modalities. Since PDAC patients do not respond well to immune checkpoint inhibitors (ICIs), novel methods for overcoming resistance are being explored. Compared to other solid tumors, the PDAC’s tumor microenvironment (TME) is unique and complex and prevents systemic agents from effectively penetrating and killing tumor cells. Radiotherapy (RT) has the potential to modulate the TME (e.g., by exposing tumor-specific antigens, recruiting, and infiltrating immune cells) and, therefore, enhance the effectiveness of targeted systemic therapies. Interestingly, combining ICI with RT and/or chemotherapy has yielded promising preclinical results which were not successful when translated into clinical trials. In this context, current standards of care need to be challenged and transformed with modern treatment techniques and novel therapeutic combinations. One way to reconcile these findings is to abandon the concept that the TME is a well-compartmented population with spatial, temporal, physical, and chemical elements acting independently. This review will focus on the most interesting advancements of RT and describe the main components of the TME and their known modulation after RT in PDAC. Furthermore, we will provide a summary of current clinical data for combinations of RT/targeted therapy (tRT) and give an overview of the most promising future directions.

## 1. Introduction

With 466,000 deaths and 496,000 cases recorded in 2020, pancreatic ductal adenocarcinoma (PDAC) is the sixth-highest cause of cancer death in both sexes [1]. The 5-year survival rate is still less than 10%, which is the lowest of all solid tumors [2,3]. PDAC may be the third leading cause of cancer death by 2025 [1]. Based on the level of vascular involvement on conventional imaging, localized PDAC is categorized into three categories: resectable (R), borderline resectable (BR), and locally advanced (LA) pancreatic cancer [4]. Treatment options are limited, and tumors frequently show limited response and develop resistance to the current treatment modalities available [5]. A complete oncological resection remains the only potentially curative option but cannot be achieved for a majority of PDAC patients [6,7]. The 5-year overall survival (OS) of resected patients is only around 20% and even for resectable, the free-margin (R0) resection rate remains suboptimal [6,7,8]. These findings and the fact that one-third of PDAC’s patients die from local progression highlight the importance of locoregional treatments such as radiotherapy (RT) and stress the need to further study innovative combinations including RT [9,10]. 

Although the role of RT is still debated in PDAC, RT is often included in the management of localized PDAC as there is currently a global shift away from upfront surgery to neoadjuvant treatments including more aggressive combination chemotherapies [11]. However, even if the improvements in patient selection are progressively currently achieved, these will not be sufficient to make a drastic improvement in the survival outcomes in PDAC. Many major obstacles are encountered with PDAC. Genomic alterations of *KRAS*, *P53*, *DCP4/SMAD4*, and *CDKN2A* genes are commonly found and are drivers of treatment resistance, and currently, none of these changes are directly druggable [3,12]. PDAC is also known to have a low tumor immunogenicity, which can be ascribed to the very low tumoral mutational burden (TMB) of PDAC, with usually a median of only four mutations identified per megabase, and only about 1% of high-TMB PDAC [6,13,14]. Moreover, tumors with high microsatellite instability (MSI-H) and mismatch repair deficiency (MMR-D) are also very rare events in PDAC (less than 1% of the cases) [13]. Many trials assessed the efficacy of immune checkpoint inhibitors (ICIs), targeted therapy, cancer vaccines, and adoptive cell transfer in PDAC [6]. However, all immune therapies have failed to improve patient survival, and contrary to other solid tumors, no practice-changing outcomes have emerged related to them [6,7,8,15]. Several researchers concluded that there must be something different about pancreatic cancer and its microenvironment, which limits its ability to respond to immunotherapy. Since these many failures, new techniques for overcoming the strong PDAC tumoral resistance as well as better therapy combinations are being investigated [15,16], including association with RT (Table 1).

Progress in PDAC treatment options has been and is still difficult; the success of future clinical trials will depend on our understanding of the complex and dynamic TME and therapy association, including RT. The first part of this review will describe the current perspective of RT in PDAC as well as the known effects and modulation induced by RT in this complex TME. The second part will provide a summary of current clinical data on RT combinations with targeted therapy (tRT). Furthermore, we will highlight the most promising clinical perspectives in PDAC tRT.

## 2. Radiotherapy Modalities and Perspectives in PDAC

The role of RT in PDAC has been intensely debated over the past 30 years. Although RT is a treatment option validated by international guidelines, the exact role of RT in PDAC remains to be further explored and validated in randomized clinical trials [37,38]. On one hand, adjuvant RT has been controversial because of inconsistent outcomes from earlier trials that used outdated prescriptions (e.g., low-dose, split-course), targeted delineation, and dosed delivery [10]. Secondly, radiosensitivity of the upper abdomen organs at risk (OAR) has also limited the radiation dose intensification. Furthermore, postsurgical hypoxia in the tumor bed may reduce the efficacy of any adjuvant treatment. On the other hand, neoadjuvant RT for localized PDAC is frequently prescribed after induction chemotherapy [10]. 

If the use of conventional chemoradiotherapy (CRT) is currently declining following the disappointing results of the available randomized phase III trials [39,40], altered dose prescriptions and new RT techniques are now showing promising oncological results associated with tolerable acute and midterm toxicity, although long-term results remain to be studied [41,42]. The importance of RT is expected to increase as new systemic treatments are developed and the local management of PDAC will become a major priority [37]. These new innovative RT techniques can also have the advantage of being more easily integrated into a combined therapy strategy than CRT [11]. 

Promising altered schemes of RT to further investigate, including innovative combination therapies, include: 

### 2.1. Hypofractionated CRT

Recently, the long-term results of the PREOPANC-1 phase III trial showed that hypofractionated neoadjuvant CRT (short induction plus concomitant gemcitabine, 36 Gy in 15 fractions) resulted in a modest improvement of median overall survival (mOS) compared with upfront surgery in R and BR PDAC (15.7 vs. 14.3 months, *p* = 0.025) [43]. Interestingly, the 5y-OS rate showed a clinically relevant improvement of 14% and the following secondary endpoints were also in favor of the hypofractionated CRT arm: disease-free survival (DFS), locoregional failure-free interval, distant metastases-free interval, and margin-negative (R0) resection rate [43]. However, a major criticism of this study was that the type of chemotherapy used was suboptimal. 

### 2.2. Hypofractionated Ablative RT (HFA-RT)

In this type of RT, the treatment delivered is hypofractionated (in 15 or 25 fractions) with concomitant chemotherapy, and the total dose delivered is sought to reach a biological equivalent dose (BED) of >70 Gy (ideally around 100 Gy) with the help of a simultaneous integrated boost (SIB) [11,44]. This target BED cut-off was associated with a better survival probability in several studies [45,46,47]. Using this technique, the Sloan Kettering group recently published the results of a nonrandomized study on a large cohort of unresectable LA PDAC. In this study, HFA-RT (67.5 Gy in 15 fractions or 75 Gy in 25 fractions, concomitant fluoropyrimidine) was delivered as a definitive treatment after modern multiagent induction chemotherapy in 119 patients. The oncological outcomes obtained were interesting with a mOS, 2y-OS rate, mDFS, and locoregional progression rate of 26.8 months, 38%, 13.2 months, and 32.8%, respectively [44]. 

### 2.3. Isotoxic High-Dose SBRT (iHD-SBRT) and MR-Guided Stereotactic Ablative RT (SMART) 

This time, the treatment is delivered in five fractions without concomitant chemotherapy. For iHD-SBRT, the dose delivered is individually tailored and maximized in order to obtain the delivery of a high BED (>70 Gy) while the dose levels to the critical gastrointestinal organs at risk (OARs) are prefixed in order to control the toxicity probability [41]. Promising oncological results were recently reported on a prospective cohort of 34 localized PDAC after induction with modern multiagent chemotherapy. The mOS, 18-month OS rate, mDFS, and 1y-local control (LC) rate were, respectively, 24.5 months, 69%, 15.6 months, and 74.1% [41]. Furthermore, in a retrospective comparison with conventional CRT, iHD-SBRT was associated with better survival, including after multivariate analysis (HR 0.39 [CI 95% 0.18–0.83], *p* = 0.014) [48]. A randomized phase II trial, the STEREOPAC trial [NCT05083247], aiming to compare FOLFIRINOX (FFX) alone versus mFFX + iHD-SBRT as neoadjuvant strategies in 256 patients with BR pancreatic cancer is ongoing [49]. 

With the use of MR-Linac, additional online daily adaptive treatments can be performed, allowing for maximal dosing to the tumor and conforming it to the GI OARs on a fraction-by-fraction basis [50]. The preliminary results of the first prospective, single-arm phase 2 study of gemcitabine doublet chemotherapy (FOLFIRINOX) followed by 50 Gy in five fractions (on a 0.35T MR-60Co or MR-Linac system) for LAPC/BRPC were presented (SMART trial, NCT03621644). Median follow-up was 16.4 months and included 136 patients from diagnosis. Less than ~2% of patients experienced acute grade 3 GI toxicity, with 1-year LC and OS being 82.9% and 93.9%, respectively [42]. The definitive results are awaited. Finally, the LAP-ABLATE trial, a randomized phase II trial, aiming to compare induction chemotherapy +/− SMART (50 Gy in five fractions) for LA PDAC, will also soon open for inclusion [51]. 

### 2.4. Intraoperative RT (IORT) and FLASH

By excluding some or all of the surrounding OARs that are radiosensitive, intraoperative radiation therapy (IORT) approaches have been developed in PDAC as another strategy allowing for dose escalation at the posterior and vascular margins of the surgical bed [52]. Usually, single doses of 10 to 20 Gy can be used alone or in addition to external beam RT. For PDAC, IORT seems to be able to offer an improvement of the LC and survival without significantly increasing the perioperative complication rates [52,53]. Several prospective trials are ongoing such as the PACER trial [54]. 

The application of radiation at extremely high dose rates (>40 Gy/s) is known as ultra-high dose rate (FLASH) RT [55]. FLASH-RT could be the best alternative for overcoming the extremely hypoxic PDAC TME while protecting the surrounding healthy intestinal tissues [56]. The FLASH effect mechanism is still not completely understood but could be an interesting technique to study in inoperable patients [55]. 

### 2.5. Low-Dose Rate RT (LDR-RT)

RT delivered at a low-dose rate (<0.30 Gy) is associated with a reaction of hyper-radiosensitivity of the cancer cells [57,58]. LDR-RT can be delivered with pulses (PDLR-RT) in order to better spare the OARs. The optimal PDLR model for PDAC was recently studied in a preclinical model and seems to be a daily radiation dose of 2 Gy divided into 10 pulses with an interval of 3.5 min and a dose rate of 100 cGy/min [57]. Several clinical studies involving LDR-RT are ongoing in PDAC such as NCT02416609, NCT00390182, and NCT04452357.

### 2.6. Spatially Fractionated RT (SFRT)

Another promising technique is SFRT, which uses an inhomogeneous dose to treat a complete or portion of a tumor [59], and the SBRT of partial tumor irradiation (SBRT-PATHY) is a particularly interesting RT perspective for PDAC [60]. SBRT-PATHY is an unconventional RT approach in which ablative doses are delivered only on the hypoxic portion of unresectable bulky tumors to spare the peritumoral tissue and further exploit the immune-stimulatory RT effects. This novel technique is in its elementary stages and clinical data are currently very limited, with very few studies conducted on bulky PDACs [61]. However, the effects of this RT on the TME and immune system have not been particularly well documented and are still under investigation. Potential developments of the SFRT concept will depend on technological development and increased knowledge of the biological processes occurring in tumor and normal tissues exposed to SFRT.

## 3. The Pancreatic Tumor Microenvironment and Its Modulation by RT

The immune landscape of cancer has emerged as a significant prognostic component; six immune subtypes have been established based on their microenvironment, genetic, and prognostic properties [62]. In comparison to other solid tumors, the PDAC’s tumor microenvironment (TME) is unique, complex, and highly dynamic due to intensive crosstalk between the different TME components, which provide many paths of resistance and tumor aggressiveness [6]. PDAC’s cells and TME also interact and influence each other, with a sophisticated interplay, which is now progressively being understood. Thus, the TME of PDAC is largely responsible for the extreme resistance to conventional and immune therapies [63]. Since the complexity of the crosstalk mechanisms is vast, developing combinations of different modulating therapies is essential to achieve a durable antitumor response. For decades, investigation on RT outcomes has mostly relied on the biological effects that radiation had on cancer cells and the normal cells of the surrounding OARs. However, RT can also affect the TME, which has significant implications for the treatment of PDAC and the establishment of innovative combination strategies. As previously exposed, RT may be delivered in a variety of techniques, doses, and fractions. However, it is still poorly known how these diverse regimens and techniques influence the TME, and intensive studies should be performed on this topic [64]. The ability to characterize the interplays and to describe the precise effects of RT on the TME will enable the development of multitargeted therapeutic strategies, which is a critical challenge. In this chapter, we will describe the main PDAC’s TME components and report their known modulation by RT, which are summarized in Figure 1.

### 3.1. Stroma and Cancer-Associated Fibroblasts (CAFs)

The PDAC stroma accounts for up to 90% of the tumor volume and is characterized by dense fibrosis, called desmoplasia, leading to an acidified and hypoxic microenvironment, a high interstitial pressure providing a drug-free sanctuary, and a physical barrier for immune cell recruitment [65,66,67]. The extracellular matrix (ECM) proteins (collagens, laminins, tenascin C, fibronectin, and hyaluronic acid) are also reported as protumorigenic and some, such as collagen I, were directly linked with increased radioresistance [68,69]. The genomic analysis of PDAC has led to the identification of two distinct stroma subtypes, normal and activated, the latter being associated with an inflammatory signature and a significantly worse prognosis [70]. To date, many preclinical and early clinical trials have focused on reducing the stroma components through the help of stroma-modulating agents [71] (e.g., Sonic hedgehog (Shh) pathway and focal adhesion kinase (FAK) inhibitors, pegylated human recombinant PH20 hyaluronidase (PEGPH20), etc.) [72,73]. In simplified preclinical models, these strategies displayed favorable results and several pathways are still investigated at the clinical level. However, several trials displayed conflicting results: an indiscriminate targeting, inducing a near depletion of the stroma, may promote PDAC aggressiveness and worsen the outcome, highlighting the fact that it is essential to take into account the inherent interplay of the TME of PDAC [71,74]. 

Despite being one of the most investigated cell types, CAFs remain the most enigmatic [75]. The CAF population originates from different cells of origin such as pancreatic stellate cells and local precursor fibroblasts [69]. Myofibroblastic CAFs (myCAFs), inflammatory CAFs (iCAFs), and antigen-presenting (apCAFs) CAFs are the three main subtypes of CAFs that are characterized in PDAC and these phenotypes appear to be interchangeable [71,76]. myCAFs are implicated in the production and remodeling of the complex ECM and are mostly identified as tumor restrictive, and iCAFs secrete multiple immune effectors (cytokines, chemokines, and growth factors) leading to an immunosuppressive PDAC TME [65,67,75]. apCAFS, the less abundant type, are associated with a prolific expression of MHC class II, but due to the lack of co-stimulatory molecules, they finally tend to counter the activation of CD4+ cytotoxic T cells [77]. As the heterogeneous CAF population harbors the capacity of both pro- and antitumorigenic functions, depleting them blindly is also not an effective way to enhance treatment response and explains the difficulty in obtaining probing clinical results [64,78]. However, targeting CAFs remains a very appealing strategy as their relative abundance in the TME and their genetic stability make them prime targets, less prone to develop resistant phenotypes compared to tumor cells [79,80]. A global and dynamic approach will almost certainly aid in comprehending the intricate roles of CAF and their interactions in PDAC as well as the development of preclinical models that can recapitulate this complexity.

####  Effects of RT

RT is known to further enhance the desmoplastic reaction in PDAC through an autocrine periostin loop [81]. In human surgical resection samples from PDAC patients treated with CRT and SBRT, an increased stromal area and collagen deposition were observed but without leading to T-cell sequestration [68,82]. The RT-induced IFN signaling, which is antitumorigenic, seems suppressed by the collagen-rich stroma in PDAC organoid culture models [68]. After exposure to RT, gene expression of collagen, integrin signaling, and FAK were induced [68]. 

Several in vitro experiments have demonstrated and established that after RT, CAFs avoid cell death and develop a senescent phenotype with decreased proliferation and migration patterns [64,83,84]. RT also inhibits CAF proliferation through persistent DNA damage [64,83]. The in vitro culture of PDAC cells with conditioned medium from irradiated CAFs led to changes in their secretome–paracrine profile while leaving the CAFs viable to still establish an active TME [64,83,85]. Also, compared to nonirradiated CAFs, those irradiated with a single dose of 5 or 10 Gy triggered an increase in the PDAC cell invasiveness [85]. In particular, the HGF-c-Met and P38 pathways were reported to be upregulated. In the end, these changes promoted tumor progression and epithelial–mesenchymal transition (EMT) through majored secretions of HGF, TGFβ, and CXCL12 [79].

However, the RT modulation is most probably not as simple as it seems, as it was demonstrated in vitro that the influence of CAFs on the tumor cell radiation response is highly variable, capable of exerting pro- and antitumoral effects, depending on the tumor cell type, culture conditions, and timing endpoint [86]. Furthermore, although RT has different effects on the irradiated tissues according to the dose regimen chosen and the time-point, the majority of in vitro studies used a single high dose and collected data very early, only a few hours/days following RT [64,83,84]. 

### 3.2. PDAC Vascular System

PDAC is a hypovascular tumor characterized by a microvascular density of microvessels with impaired integrity and which are poorly perfused due to desmoplasia-induced collapse, resulting in a hypoxic TME [87,88]. Hypoxia, particularly through the HIF pathway, which supports the adaptation of tumor cells in an oxygen-deficient TME, is an important factor of resistance to chemotherapy/RT [89,90]. Vascularization in PDAC emerges from various types of angiogenic and nonangiogenic pathways [91]. This could explain the poor efficacy of antiangiogenic therapies in PDAC. Tumor cells and the other components of the TME globally support and use this deficient intratumoral vasculature to promote PDAC cell migration [91]. Furthermore, the presence of basal microvilli on the microvessels allows the PDAC cells to maintain a high glucose uptake despite the deficient vascular system [92]. 

#### Effects of RT

RT has been shown to induce endothelial cell dysfunction and apoptosis, thereby leading to further hypoperfusion and reducing vessel density [93]. This was also observed in PDAC mice models where RT (1 × 20 Gy) resulted in a significantly lower number of microvessels and an increase in vascular endothelial growth factor (VEGF) [94], a proangiogenic cytokine. However, in human samples after SBRT (5 × 5 Gy), no difference in the number of endothelial cells or vessel area/density was found. This study suggested that SBRT does not affect the PDAC vasculature and is even capable of improving CD8+ T-cell migration from the vessels [82]. 

### 3.3. Immune Cells

The PDAC’s TME is made up of a variety of immune cells that can influence results in either a positive or negative way; therefore, these cells must be fully explored before designing treatment modalities [95]. Tumor-associated macrophages (TAMs), tumor-associated neutrophils (TANs), regulatory T cells (Tregs), and myeloid-derived suppressor cells (MDSCs) are among the prominent cells in the TME [73,96]. 

#### 3.3.1. Tumor-Associated Macrophages (TAMs)

TAMs can be differentiated into different subgroups harboring different functions according to their state of polarization, which is also a dynamic process. In PDAC, most of the TAMs are inclined into the side of the M2 polarization type associated with an anti-inflammatory and protumorigenic phenotype [97]. The M2-type TAMs secrete various immunosuppressive cytokines into the TME such as IL-10, TGF-β, and CCL2 [97,98]. Through the regulation of matrix proteases (e.g., ADAM8 and MMP9) and chemokines (e.g., CXCL8 and CXCL12), M2-type TAMs aggravate angiogenesis and tumoral migration [99,100]. These TAMs are also known to foster desmoplasia, EMT, tumor invasion, and chemotherapy resistance [101]. 

Again, although various tactics to target TAMs (through depletion, impaired recruitment, and reprogramming) were promising in preclinical studies, their effectiveness is difficult to realize in clinical trials [101]. 

##### Effects of RT

Following the exposition of PDAC to RT in mice models and compared to control mice, a higher tumoral infiltration by TAMs was reported [102]. Furthermore, an increased proportion of these TAMs presented an M2-like phenotype, which led to the deactivation of the T-cell-mediated antitumor responses [102]. After RT, levels of TGF-β and IL-10, major cytokines secreted by M2-TAMs, were also reported to be significantly increased [103,104]. The recruitment of TAMs and their RT-induced reprograming appeared to be driven by the upregulation of macrophage colony-stimulating factor 1 (MCSF) in PDAC cells three days after RT. The use of neutralizing antibodies against MCSF prevented the shift of phenotypes after treatment by RT and slowed tumor growth [102]. It is interesting to note that in this study, different types of RT dose prescriptions were used (2–12 Gy in a single fraction, three times 6 Gy at 48-h intervals, and a single dose of gemcitabine followed by the delivery of 12 Gy a day later), with the same effects on the macrophage population. 

Another important origin of this increased TAMs infiltration post-RT was identified in a preclinical study. Ablative RT (a single dose of 20 Gy) was reported to induce an increase in the secretion of CCL2 by PDAC cells, which acts as a mechanism of RT resistance [104,105]. CCL2 is a chemoattractant for the myeloid cells from the bone marrow. In consequence, TAM and inflammatory macrophages/monocytes were increased 3 days postirradiation in in vitro and mouse models [105]. In fine, this RT-stress-induced CCL2 secretion inhibits the efficacy of FFX and ablative RT in mice. The use of the CCL2 selective blockade restored the sensitivity to chemo/radiotherapy by impairing macrophage recruitment [105]. It is interesting to note that the inhibition of CCL2 had little impact on tumor growth and vascularity if not used in combination with RT [105,106]. In another recent preclinical study, different combinations of dual antagonists of CCR2/CCR5, anti-PD-1 and PDAC cell vaccine (GVAX), and SBRT (3 × 8 Gy) were investigated in PDAC mouse models [107]. Compared to the PDAC cell vaccine, SBRT was the most adequate T-cell priming agent identified in this study. The best timing and combination identified was SBRT followed by anti-PD-1, then prolonged by the dual CCR2/5 inhibitor. This triple combination displayed the best antitumoral results and was capable of enhancing intratumoral effector/memory T-cell infiltration while suppressing the regulatory T-cell (Treg), M2-like TAMs, and MDSC populations, notably by establishing a more favorable expression of chemokines. 

#### 3.3.2. Myeloid-Derived Suppressor Cells (MDSCs)

In PDAC, the MDSCs are a heterogeneous cell population involved in the maintenance of an immunosuppressive state [108]. Intensive crosstalk exists notably between them, TAMs, Tregs, and CAFs. The most frequent subtype in PDAC is the granulocytic (G-MDSCs, 70–80%) followed by the monocytic phenotype (M-MDSCs, 20–30%) [109]. Through the STAT3-TANs pathway, G-MDSCs lead to the post-translational modification of T-cell receptors and their unresponsiveness. M-MDSCs also suppress the T-cell response and promote T-cell apoptosis but this time through the STAT1/NO axis [109,110]. Both subtypes also exhibit increased activity of arginase 1, which also leads to T-cell suppression [109,110]. 

##### Effects of RT

In PDAC mouse models, RT (a single dose of 8 Gy) was reported to enhance the entire population of MDSCs (both G- and M-MDSC subtypes) [104]. In particular, the expression of phosphorylated (p)STAT3 was increased on G-MDSCs and neutrophils. Inhibition of STAT3 allowed the reversal of RT-induced immunosuppression, decrease the load of the ECM, and improve tumor response to RT [104]. Furthermore, the immunosuppressive activity of MDSCs appears to be reinforced after RT via the support of the Warburg effect [103]. This effect, which promotes DNA damage repair and leads to radioresistance, is defined as a tumoral metabolism characterized by enhanced glycolysis and accelerated lactate production rates [103,111]. The use of a lactate dehydrogenase A (LDHA) blockade inhibited tumor growth and MDSCs activation as well as antitumor immunity after RT [103]. With the use of another SBRT dose prescription (5 × 6 Gy) in a mouse model, no major changes in the myeloid cell populations were observed [68], highlighting the fact that further well-conducted studies are required to precisely understand the immunomodulation induced by different dose prescriptions. 

#### 3.3.3. Tumor-Associated Neutrophils (TANs)

TANs participate in the immunosuppressive role of the TME in PDAC. Similar to TAMs, TANs are often classified into two simplified opposite phenotypes, the N1 (antitumorigenic) and N2-TANs (protumorigenic and immunosuppressive), despite a more mixed reality [112,113]. The N2-like phenotype is more predominant in the recruited TANs in PDAC. Therefore, TANs were shown to significantly promote migration and invasiveness of PDAC through various mechanisms [112]. One of these mechanisms, specific to TAN, is the induction of neutrophil extracellular traps (NETs) consisting of a spider web-like extracellular release of decondensed DNA and proteins such as the neutrophil elastase. Originally, NETs are a defense mechanism designed to trap and kill bacteria [114]; however, when induced by PDAC cells in the TME, these NETs enhance tumor aggressiveness and EMT and play a pivotal role in the metastatic cascade [115,116]. Modulation of TANs is also a promising strategy to explore in PDAC, as recently illustrated [117]. 

##### Effect of RT

Many studies have focused on the prognostic impact of blood’s neutrophil-to-lymphocyte ratio (NLR). Pre- and post-RT (CRT and SBRT) NLRs were analyzed in several studies and correlated to the oncological outcomes; a high NLR post-RT was associated with poor outcomes [118,119,120,121]. The RT-induced modulation of TANs in the TME of PDAC is until now poorly investigated. SBRT (3 × 8 Gy) was suggested to activate neutrophil recruitment and to promote their polarization in the N1-like phenotype in lung cancer, but this was not investigated in PDAC [122]. 

#### 3.3.4. Tumor-Infiltrating Lymphocytes (TILs)

In PDAC, the vast majority of tumor-infiltrating lymphocytes (TILs) promote carcinogenesis, while the effector T cells are scarce [12,123]. TILs encompass many different subsets such as the cytotoxic (CD8+) T cells, Treg, memory T cells, and natural killer (NK) cells [124]. As previously exposed, effector T-cell infiltration and activation is made difficult by the inherent desmoplasia, as well as indirectly by many immune-suppressing cells, cytokines, and metabolites that contribute to resistance and progression [73,82,123,125]. Moreover, the PDAC cells can downregulate the expression of MHC class molecules to mitigate the antigen presentation to CD8+ T cells and express, in association with MDSCs, the immune checkpoint molecule PD-L1 mediating T-cell exhaustion and cell death [126,127]. The localization and characterization of intratumoral T cells in untreated PDAC have been investigated [96]. Increased infiltration of global T cells was independently related to prolonged OS in multivariate analysis [96]. Interestingly, cytotoxic CD8+ T cells adjacent to the cancer cell (20 µm) were strongly related to longer OS [96]. This result emphasizes the importance of “cell–cell” interaction in cytotoxic activity. However, in reality, T cells are more complex to investigate, since their functional role is challenging to quantify and, therefore, not commonly assessed [128]. CD8+CXCR5+ T cells, a subgroup of cytotoxic cells identified in PDAC with high functionality, were associated with survival and were particularly responsive to PD-1 and TIM-3 blockades [129]. Furthermore, in the tumor of treatment-naïve long-term PDAC survivors, the presence of a high neoantigen number and an abundant CD8+ T-cell infiltration have been reported [130]. Interestingly, the high-quality neoantigens were found to be lost in the metastatic samples [130].

Treg cells are an important subpopulation of CD4+ T cells in PDAC [124]. They take part in the suppression of the effector T-cell response through various mechanisms [131] such as the direct elimination of effector T cells by granzymes and perforins or the secretion of immunosuppressive cytokines (TGF-β, IL-10, etc.) [132]. However, Treg remains essential for the maintenance of the immune tolerance of the body and they also share, in common with the other types of T cells, many molecular signaling pathways. Therefore, these facts make the Treg in the TME difficult to be specifically targeted [132]. 

B cells (CD20+) compose a modest population of the TME, with a B-cell/T-cell ratio of only ≃ 1:6 [128]. However, it appears that the organization of B cells, rather than their number, is relevant. The existence of tertiary lymphoid structures (TLS) in the (peri)tumoral stroma was correlated with a greater OS and was also associated with a more favorable immune cells infiltration profile [128,133]. 

The role of NK (CD56+) in PDAC remains poorly understood compared to the other types of lymphocytes. The NK heterogeneous population is composed of two main subtypes, immunomodulatory and cytotoxic NK [134]. PDAC cells have developed several methods to evade their cytotoxic effect, notably by the downregulation of activating receptors and exposure to MMP9 and IDO [134,135,136]. 

##### Effects of RT 

RT functions as an in-situ vaccine, producing DNA damage and radiation-induced neoantigens that could stimulate T-cell infiltration and a robust tumor-specific immune response [123]. Furthermore, RT can upregulate the MHC-I expression on the tumor surface, allowing for a better presentation of tumor-specific peptides and increasing the PDAC tumor visibility to cytotoxic T cells [5]. Nevertheless, the neoantigen quality, rather than the quantity, ascribes greater immunogenicity [130]. It is therefore crucial to investigate the neoantigens’ quality following neoadjuvant therapy and select the combination that would produce the highest quality, which can subsequently be effectively exploited. 

In blood samples of PDAC patients treated with neoadjuvant CRT (50.4 Gy in 28 fractions) or SBRT (3 × 10 Gy), circulating lymphocytes and cytokine signaling were only preserved in patients treated with SBRT. Therefore, SBRT seems to be a better partner for ICI combinations than CRT [137]. 

In PDAC preclinical studies, SBRT has been shown to effectively recruit and stimulate cytotoxic T cells [95]. In samples of SBRT-treated human PDAC, preoperative SBRT did not significantly decrease the intratumoral T-cell distribution of CD8+/CD4+ [68,82]. Furthermore, CD8+ T cells were found considerably less near (within 40 µm) the vessels, suggesting that SBRT contributed to their infiltration [82]. Low-BED SBRT was also found to reduce the number and area of TLS [82]. However, the remaining TLS contained fewer immunosuppressive cells (such as Treg and TAMs) and were found closer to cancer cells for supporting immune effects [82]. This might be explained by the fact that SBRT could provide dynamic and interactive TLS remodulation. In addition, although most of the immunosuppressive cell density was preserved after SBRT, the Treg population was found significantly decreased in this study [82]. Finally, a significant increase in PD1/PD-L1 expression was observed following SBRT [82].

If SBRT was shown to indeed induce immunogenic cell death (ICD) through the release of DAMPs (HMGB1 and HSP70) in human PDAC clinical samples, modifications of the dose and timing of SBRT and FFX administration demonstrated to have a significant influence on the immune response and ICD in murine PDAC models [82,95]. In a preclinical study, SBRT (4 × 6 Gy) associated with concomitant FFX has been found to be the optimal sequence to obtain the highest level of ICD compared to sequential treatment [95]. The concurrent administration enhanced the local antigen presentation, maturation of dendritic cells, and systemic antigen presentation also in nontumor-draining lymph nodes and the spleen [95]. 

In another study, the gene expression patterns and spatial TILs distribution of 4 × 6 samples of human PDAC treated with, respectively, surgery alone or neoadjuvant FFX +/− RT (conventional CRT or SBRT) followed by surgery, were analyzed with the GeoMX^®^ platform [138]. The addition of RT induced more profound modifications in gene expression than chemotherapy alone. Neoadjuvant therapies were linked to durable significant variation in the quantity of various pan-immune cell markers, implying a long-term reconfiguration of the TME [138]. No differentially expressed genes were identified between the SBRT and CRT groups. FFX alone was not found to depopulate the TILs and was associated with higher PD-L1 and granzyme B expression [138]. On the other hand, the combination of FOLFIRINOX + SBRT was associated with a significantly lower expression of *PD-1*, *VISTA*, *AKT*, *ß-catenin*, *STAT3*, and T cells [138]. B cells were significantly lowered after FOLFIRINOX +/− SBRT [138]. It is noteworthy that, (i) the number of samples studied was low, (ii) the number of induction FFX cycles received was lower than what is usually used in daily practice (mean number of cycles: 2), (iii) the time between the end of the neoadjuvant therapy and surgery was significantly higher for the RT groups (mean: 9.4–17 vs. 6.6 weeks for FFX alone), which could result in greater dead cells and fibrosis, and (iv) unfortunately, no RT parameters (dose, fractionation, or techniques) were provided. 

Recently, a study investigated the immunomodulation effect of IORT in PDAC, which is poorly known [139]. Following the IORT treatment of PDAC (10 Gy at a depth of 5 mm into the tumor bed), elevated levels of several key cytokines were observed in the peritoneal fluids compared to patients who underwent surgery alone, such as TGF-β, INF-γ, IL-15, and PDGF-BB (associated with the PI3K/SMAD pathway). 

Modifications to the immune cells’ blood populations were also observed as an increase in cytotoxic and helper T cells, NK cells, and IFN- γ production, while the Treg cells remained low [139]. 

### 3.4. Others 

#### 3.4.1. GMP-AMP synthase (cGAS)-stimulator of interferon genes (STING) pathway

In recent years, new therapeutic possibilities have emerged with the finding of a novel target protein that might induce the antitumor response [140]. In this light, the cyclic GMP-AMP synthase (cGAS)-stimulator of interferon genes (STING) pathway has been investigated in preclinical PDAC models and has been found to be a key player [140,141,142,143]. STING ligands in PDAC can affect both cancer cells and TILs, influencing the production of inflammatory cytokines and type I interferons (IFN-α, -β) leading to the facilitation of adaptive immune responses to destroy the tumor cells [140,143,144]. Unlike other gastrointestinal tumors, STING and cGAS expression were found to be rather preserved [142]. PDAC cGAS/STING + tumors were strongly associated with better OS and CD8+ T-cell infiltration. Conversely, in PDAC tumors without expression of cGAS/STING, the cytotoxic CD8+ T-cell infiltration was located farther away in the tumor periphery, and the TILs’ stromal infiltration was inhibited [142]. Therefore, cGAS/STING might be an important signal for immune cells to overcome and infiltrate the stroma barrier, and precisely, RT is known to be able to enhance this pathway under a certain dose range (8–12 Gy) [142,145,146].

A preclinical study analyzed the combination of SBRT (10 Gy) with a STING agonist (RR-CDG) in PDAC models [143]. This combination generated systemic immune responses capable of controlling the distant disease. First, TNF-α secretion, which induced hemorrhage necrosis, contributed to the early (+6 h) tumor control, and secondly, the late control (+7 days) was shown to be CD8+ T-cell dependent [143]. 

However, the STING pathway repercussion on other PDAC TME elements, its long-term activation, and its polymorphisms in humans, require additional investigation [140].

#### 3.4.2. Microbiome 

The pancreas was originally considered to be a sterile organ, but the discovery of bacteria and fungus within pancreatic tumors has recently been shown to play an important role in the TME effect [147,148,149,150]. Bacterial eradication was linked to TME reprogramming, including a reduction in myeloid-derived suppressors, an increase in M1-TAMs development-boosting TH1 differentiation (antitumorigenic), and CD8+ T-cell activation [150]. These data, however, are limited to associations rather than causal relationships [148]. RT can disrupt the microbiome, leading to dysbiosis, and such changes can impact anticancer treatment effectiveness in a bidirectional way [151]. Crosstalk between the microbiome and the TME may modify radiosensitivity and potentially transform “cold” tumors into “hot” tumors [151]. Although not yet mature enough, targeting the microbiome may have therapeutic relevance, and relationships between the microbiome and the TME, as well as their interactions with cancer therapy, should be investigated.

## 4. Clinical Targeted Treatment Combinations including RT and Perspectives

### 4.1. Combination of Radiotherapy and Targeted Therapy in PDAC (tRT)

In recent years, immunological checkpoint inhibitors (ICIs) as well as various other targeted therapies and their combinations with RT (tRT) have gained momentum resulting in a tremendous development of this topic [5,8,152]. RT, in addition to the direct damage to irradiated tumor cells, is now recognized as an immune-priming agent of choice [72,82,146,153]. tRT, particularly with ICIs, demonstrated preclinical synergistic effects which were translated successfully at the clinical level in other cancers [153,154,155], leading to the establishment of new treatment standards. However, this was not the case for PDAC [156] and, as previously described, these disappointing results have pushed researchers to further investigate its TME to find innovative ways to overcome the intertwined resistance pathways. 

With RT, the equilibrium of immunostimulatory and immunosuppressive responses should be seen as a double-edged sword, particularly in PDAC. As we previously reported, following RT in PDAC, the recruitment of immune suppressive cells (e.g., MDSCs, Tregs, TAMs, etc.), as well as their reprogramming (M2-like TAMs), counterbalance the antigen exposure and the antitumorigenic immune cell’s recruitment [102,157]. One of the most challenging tasks will be to reestablish and boost T-cell attractiveness throughout the PDAC’s TME and their activation into the TLS, which will require a multitargeted strategy. Currently, the published results of clinical trials involving tRT (usually a combination of SBRT with single-agent or dual ICIs) have been globally disappointing, as summarized in Table 1. Another issue is that the optimal dose scheme and RT timing for inducing the best immunological response are still not well known in general and particularly for PDAC [153]. This is illustrated by the wide variety of dose schemes used throughout the different trials. These crucial parameters must be thoroughly investigated in PDAC (pre)clinical trials. The main ongoing trials with tRT are also listed in Table 1. 

Other methods are also under study to overcome the related TME issues in PDAC by improving treatment delivery/efficacy, such as nanoparticles (NP) [158,159]. Recent successful clinical results have already been obtained in PDAC for the following chemotherapy nanocarriers: nab-paclitaxel (Abraxane^®^) associated with gemcitabine and the nanoliposomal irinotecan (Onivyde^®^) associated with fluorouracil and folinic acid [160,161,162]. The NP-based delivery of targeted therapy is an interesting strategy that is under development for PDAC and other cancers [163,164,165]. NP can also be used as local radiosensitizers, allowing increased DNA damage and the production of reactive oxygen species (ROS) [166]. Several types of NP-radiosensitizers are studied, including in PDAC. A randomized phase I/II clinical trial is currently recruiting in the United Sates for localized lung tumors and LA PDAC (n = 100), aiming to compare the treatment efficacy of MR-guided SBRT +/−, a gadolinium-based NP radiosensitizer (NCT04789486) [167]. Combinations of NP-based therapy with RT should be further investigated in the future. 

### 4.2. Selected Promising Clinical Perspectives of tRT in PDAC

Figure 2 resumes the selected promising tRT in PDAC.

#### 4.2.1. Targeting the CCL2 Axis

As the CCL2 axis is an important pathway leading to the recruitment of TAMs after RT and the combination of CCL2 inhibition with RT has already shown interesting results in a preclinical study [105], all of these make this strategy a prime candidate to boost RT. The blockade of CCR2, the receptor of CCL2, was investigated in a phase I study with a safe profile in combination with FOLFIRINOX only for 47 BR/LA patients [168]. A phase I/II trial is currently ongoing, studying the combination of anti-PD-1 (Nivolumab) + anti CCR2/5 (BMS-813160) +/− GVAX after an induction by FFX followed by SBRT (5 × 6.6 Gy) in LA PDAC patients. As the preclinical results of these combinations were particularly promising, the results of this study should be followed with interest [169]. 

#### 4.2.2. Targeting the TGF-β Axis

The inhibition of TGF-β is an interesting tRT to explore as it is now well established that TGF-β is a key driver of immune evasion in PDAC [98] and levels of TGF-β are increased after RT. Preclinical results of the inhibition of TGF-β were promising as it has been shown to increase responses to ICIs [98]. Currently, two TGF-β inhibitors have been tested clinically in PDAC, galunisertib and bintrafusp alfa (M7824). The first one was recently explored in combination with anti-PD-L1 durvalumab in a phase 1b study including 32 metastatic PDAC patients. Unfortunately, the clinical efficacy of the combination was very limited (only one patient with a partial response and seven with stable disease; mOS of 5.7 months) although rather well tolerated. Regarding Bintrafusp alfa, a bifunctional fusion protein composed of antibodies against PD-L1 and TGF-β, although the preclinical results with RT were promising [170], the clinical results available are still limited but already appear disappointing. In a recent phase I trial, five heavily pretreated metastatic or LA PDAC patients were exposed to this molecule and only one, with MSI-high, showed a partial response [171]. When combined with gemcitabine, bintrafusp alfa is too toxic; a phase I trial was recently terminated due to the occurrence of grade 3/4 treatment-related adverse events in all the patients (anemia, thrombocytopenia, GI hemorrhage, among others), including one death after treatment-induced hepatitis (NCT03451773) [172,173]. A phase I/II trial was recently designed to study the combination of bintrafusp alfa, immunocytokine NHS-IL12, and SBRT for BR or LA PDAC patients. Unfortunately, the study was closed to accrual in the phase I part before the addition of SBRT due to the worsening risk/benefit ratio in metastatic PDAC patients receiving bintrafusp alfa (NCT04327986) [173]. Other TGF-β inhibitors are currently being explored, without RT, in early clinical trials for PDAC (NCT04624217, NCT04390763); their results are awaited and their combination with RT should be investigated. 

#### 4.2.3. Targeting the Angiotensin Axis

The renin–angiotensin axis is long known to regulate renal and cardiovascular homeostasis but is also implicated in cancer cell proliferation, metabolism, and tumor growth [174,175]. Activation of this axis in CAFs was shown to enhance desmoplasia through the TGF-β pathway. Angiotensin I-II receptor blockers are well-known and inexpensive drugs, such as losartan and olmesartan, which have been repurposed in cancer animal models with promising results. Losartan has the potential to reduce the load of the ECM by reducing collagen/hyaluronan production leading to improved functional microvasculature in PDAC [176,177] and better drug delivery. Interestingly, a retrospective study was carried out in the United States, evaluating 794 resected or metastatic PDAC patients according to their use of angiotensin pathway inhibitors. Resected patients (n = 299) chronically taking angiotensin pathway inhibitors had superior mOS than nonusers (36.3 vs. 19.3 months, *p* = 0.011), including after multivariate and propensity score analysis [178]. The genomic expression of user-patients highlighted a normalized ECM, less expression of genes involved in PDAC progression as well as an increase in genes related to T-cell and antigen-presenting cell activity [178]. Therefore, a phase II trial was designed to investigate a total neoadjuvant therapy associating FOLFIRINOX with losartan followed by RT in 49 LA PDAC patients [24]. In this study, a personalized RT method was employed: if the tumor was resectable, short-course RT with protons (25 GyE in five fractions) or photon (30 Gy in ten fractions) was utilized; otherwise, long-course CRT was employed (50.4 Gy with a vascular boost to 58.8 Gy). Furthermore, an IORT boost was permitted (10–15 Gy). The primary endpoint was met with an R0 RR of 66%. Moreover, the survival outcomes were promising with a mOS and mPFS of 31.4 and 17.5 months, respectively. Circulating biomarker evaluation was also performed and losartan was indeed associated with a significant decrease in TGF-ß levels [24]. This led to the development in the United States of a larger randomized phase II trial for BR/LA PDAC patients with four arms: (i) neoadjuvant FFX (eight cycles) followed by SBRT and surgical resection; (ii) neoadjuvant FFX (eight cycles) plus losartan followed by SBRT plus losartan, then surgical resection and 6 months of losartan; (iii) neoadjuvant FFX (eight cycles) plus losartan followed by SBRT plus losartan and nivolumab, then surgical resection and 6 months of losartan and nivolumab; and (iv) neoadjuvant FFX (eight cycles) followed by SBRT plus nivolumab, then surgical resection and 6 months of nivolumab (NCT03563248). To date, the results of this clinical trial are pending.

#### 4.2.4. Targeting the DNA Damage Response Axis

One way to reduce the radioresistance induced by the activation of DNA damage response (DDR) is to use an inhibitor of Wee1, a tyrosine kinase acting as a key regulator of the cell cycle G2 checkpoints [23,179,180] and replication stress. PDAC cells may be particularly sensitive to Wee1 inhibitors due to the association of their RAS mutations with an increased baseline replication stress [181]. By inhibiting Wee1, the replication stress further majors, and the G2 checkpoint is abrogated, which leads the PDAC cells to undergo mitosis before repairing the RT-induced DNA damage, causing their death [23,182,183]. In LA PDAC, neoadjuvant gemcitabine followed by CRT (52.5 Gy in 25 fractions) was delivered in association with a Wee1 inhibitor (Adavosertib, AZD1775) in a phase I trial [23]. This combination was well tolerated, and with a median follow-up of 15 months, the mOS was 21.7 months. Another Wee1 inhibitor (ZN-c3) is also currently being tested in several solid tumors, including PDAC (NCT05431582). This promising axis should be studied further in combination with modern therapies such as SBRT and FOLFIRINOX. 

#### 4.2.5. Targeting the FAK Axis

By modulating the ECM, proinflammatory pathway, and CAFs, focal adhesion kinase (FAK) inhibitors are an interesting axis to explore [68,184,185]. FAKs are nonreceptor tyrosine kinases that are hyperactivated in PDAC and associated with cell migration, proliferation, and poor survival [186,187]. FAK signaling is also a key driver of the fibrotic and immunosuppressive TME of PDAC [188]. FAK inhibition was demonstrated to render PDAC responsive to ICIs in mouse models [68,188]. Using in vitro and in vivo PDAC systems, a FAK blockade led to an improved SBRT efficacy (5 × 6 Gy), T-cell priming, and long-term survival in mouse models [68]. These promising results led to the development of an ongoing randomized phase II trial, evaluating the role of defactinib (FAK inhibitor) in LA PDAC patients treated with neoadjuvant chemotherapy followed by SBRT (5 × 10 Gy) (NCT04331041). The results of this trial should be followed carefully. 

## 5. Conclusions 

PDAC remains one of the deadliest solid tumors and very few impacting therapeutic advances have been made in the last decades. Furthermore, the level of radio-, chemo-, and targeted therapy resistance is high, mainly due to the particularly immunosuppressive TME of PDAC. Efforts to better understand the complex interplay of the different components of this TME as well as the precise modulations induced by the different schemes of RT available should be strongly encouraged. Finding innovative ways to modulate the PDAC’s TME interplay could be key to improving treatment efficacy, and modern RT should be seen as an ally of choice. Nevertheless, preclinical results must be confirmed when translated clinically but, unfortunately, this is rarely the case in PDAC. For this purpose, it is crucial to invest efforts to design more adequate preclinical models, able to better reflect the real and hard complexity of PDAC. However, several promising combinations of targeted therapies with RT are already on their way at the clinical level and may be able, in the future, to turn the tables in the management of pancreatic cancer. 

## Figures and Tables

**Figure 1 cancers-15-00768-f001:**
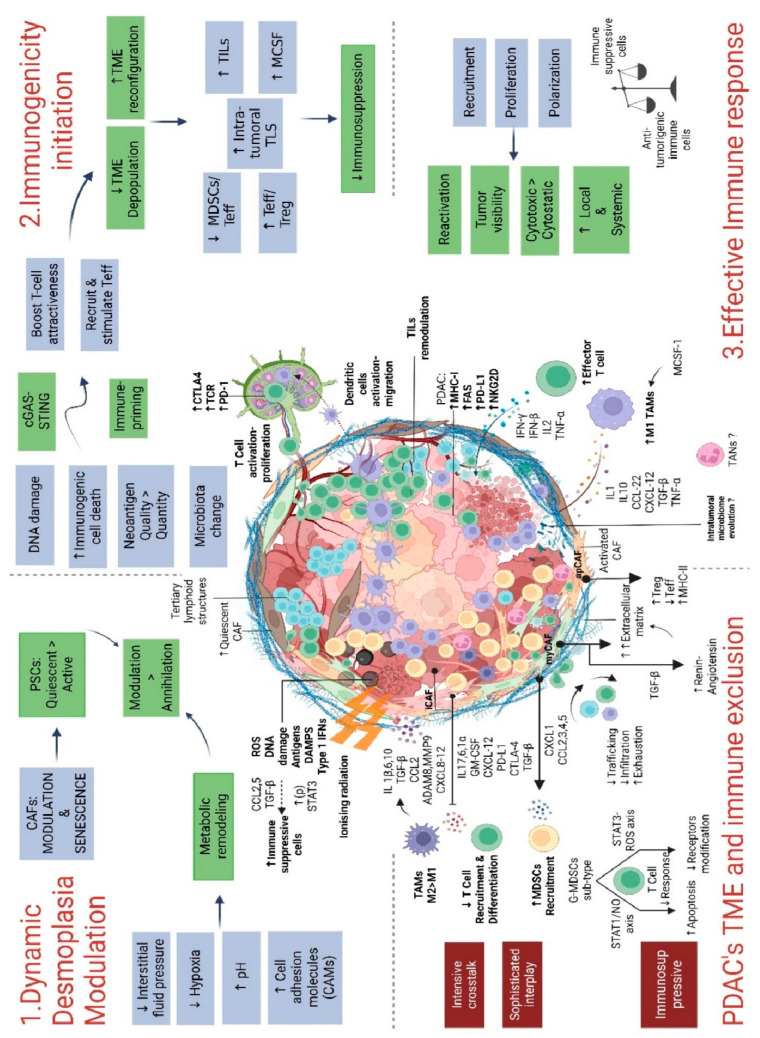
An overview of how radiotherapy affects the PDAC’s TME. Abbreviation: PDAC: pancreatic ductal adenocarcinomas; TME: tumor microenvironment; MDSCs: myeloid-derived suppressor cells; G-MDSCs: granulocytic myeloid-derived suppressor cells; MHC: major histocompatibility complex; TAMs: tumor-associated macrophages; TANs: tumor-associated neutrophils; CAF: cancer-associated fibroblasts; myCAFs: myofibroblastic cancer-associated fibroblasts; iCAFs: inflammatory cancer-associated fibroblasts; apCAFs: antigen-presenting cancer-associated fibroblasts; ROS: reactive oxygen species; DAMPs: damage-associated molecular patterns; TLS: tertiary lymphoid structures; TILs: tumor-infiltrating lymphocytes; Teff: effector T cells; Treg: regulatory T cells; MCSF: macrophage colony-stimulating factor.

**Figure 2 cancers-15-00768-f002:**
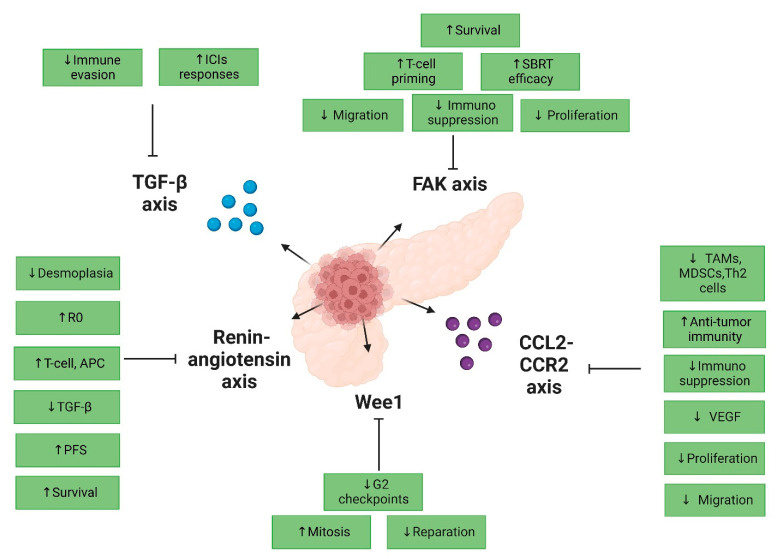
Selected promising clinical perspectives of tRT in PDAC. Abbreviation: R0: R0 resection; APC: antigen-presenting cell; PFS: progression-free survival; SBRT: stereotactic body radiotherapy; TAMs: tumor-associated macrophages; MDSCs: myeloid-derived suppressor cells; VEGF: vascular endothelial growth factor; ↑: increased; ↓: decreased.

**Table 1 cancers-15-00768-t001:** Main clinical trials with tRT combinations in PDAC.

Reference	Phase	Setting	N	Systemic Agents	RT Scheme	RT Timing/Immunotherapy	Primary Endpoint	Note
**Clinical trials with published results**								
**Lutz et al. 2011** [17]	II	Adjuvant	60	GVAX (cancer vaccine)	28 × 1.8 Gy (+ 5-FU)	After 1st dose	mDFS: 17.3 mo.	
**Picozzi et al. 2011** **(ACOSOG Z05031)** [18]	II	Adjuvant	89	Cisplatin-5FU + INFα2b	28 × 1.8 Gy	At cycle 1	OS at 18 mo.: 69%	Toxicity failure (95% grade ≥3)
**Crane et al. 2011** [19]	II	Unresectable LA	69	Gemcitabine + Oxaliplatin + Cetuximab	28 × 1.8 Gy (+capecitabine)	After 4 cycles	1y-OS: 66%	
**Hardacre et al. 2013** [20]	II	Adjuvant	70	Gemcitabine + Algenpantucel-L (cancer vaccine)	28 × 1.8 Gy (+ 5-FU)	At 3rd dose	1y-DFS: 62%	
**Chan et al. 2016** [21]	I	Neoadjuvant R/BR/LA	21	Vorinostat (HDAC inhibitor)	10 × 3 Gy	Day 1	MTD	mOS: 13 mo.
**Brar et al. 2019 (abstract)** [22] **NCT02311361**	Ib/II	Unresectable + nonresponder to chemotherapy	51/65	Durvalumab +/− Tremelimumab +/−	1 × 8 Gy or 5 × 5 Gy	Day 1	Feasibility/Safety	Preliminary results
**Cuneo et al. 2019** [23]	I	Neoadjuvant LA	34	Gemcitabine + Adavosertib (Wee1 inhibitor)	25 × 2.1 Gy	At cycle 2	MTD	mOS: 21.7 mo.
**Murphy et al. 2019** [24]	II	Neoadjuvant LA	49	FFX + Losartan	5 × 5 GyE (protons) or 10 × 3 Gy or 28 × 1.8 Gy (+capecitabine)	After cycle 8	R0 RR: 61%	mOS: 31.4 mo. mPFS: 17.5 mo.
**Lin et al. 2019** [25]	Ib/II	Neoadjuvant LA	11	Gemcitabine-5FU-leucovorin + Oregovomab (anti-CA-125) + Nelfinavir mesylate	5 × 8 Gy	At week 11	mPFS: 8.6 mo.	Closed prematurely due to outdated chemotherapy
**Tuli et al. 2019 (abstract)** [26] **NCT03245541**	Ib/II	Neoadjuvant LA	18/30	Gem-Np + Durvalumab	5 × 6.6 Gy	Day 8	Safety, PFS, RR	Preliminary results – mPFS: 14 mo., RR: 50%
**Xie et al. 2020** [27]	I	M+ (2nd line)	59	Durvalumab + Tremelimumab	1 × 8 Gy or 5 × 5 Gy	Day 1 Day -3 to +1	Safety	ORR: 5.1%
**Parikh et al. 2021** [28]	II	M+ (MSS)	25	Nivolumab + Ipilimumab	3 × 8 Gy (photons or protons)	At cycle 2	DCR: 20%	ORR:12%
**Poklepovic et al. 2021 (abstract)** [29] **NCT02349867**	I	Neoadjuvant R/BR/LA	22	Gemcitabine + Sorafenib + Vorinostat	28 × 1.8 Gy	Concurrent	Recommended dose	Preliminary results
**Rahma et al. 2021 (abstract)** [30]	Ib/II	Neoadjuvant R/BR	37	Pembrolizumab +/− FFX	28 × 1.8 Gy (+capecitabine)	Concurrent	Safety/TILs density	Preliminary results–mOS: 27.8 mo. TILs: No difference
**Lee et al. 2021 (abstract)** [31] **NCT02648282**	II	Neoadjuvant LA	58	FFX or Gem-Np + GVAX + Pembrolizumab	5 × 6.6 Gy	In cycle 2, concurrent	DMFS: 9.7 mo. (NS)	
**Haldanarson et al. 2022** [32] **(N064A Alliance)** **NCT00601627**	II	Unresectable LA	52	Panitumumab (EGFR inhibitor) + Gemcitabine	28 × 1.8 Gy (+ 5FU)	Day 1	1 y-OS rate: 50%	OS and toxicity failure (88% grade ≥3)
**Zhu et al. 2022** [33]	II	Post-op local recurrence (KRAS mut and PD-L1+)	85	Pembrolizumab + Trametinib	5 × 7–8 Gy	2 weeks before ICIs	mOS: 14.9 mo.	
**Chen et al. 2022** [34] **CHECKPAC**	II	Refractory M+	84	Nivolumab +/− Ipilimumab	1 × 15 Gy (single lesion)	Day 1	CBR: 37.2% for triple combination	ORR: 14% for the triple combination mOS: 3.8 mo.
**Lierman et al. 2022** [35] **(PARC)**	rII	Inoperable PDAC	68	Gemcitabine +/− Cetuximab	25 × 2.16 Gy	After 1 week	mPFS: 6.8 mo.	mOS: 14.2 mo. (NS)
**Hewitt et al. 2022** [36]	III	Neoadjuvant BR/LA	303	FFX or Gem-Np +/− Algenpantucel-L	28 × 1.8 Gy (+ 5-FU or capecitabine)	After 6 doses	mOS: 14.3 mo. (vs. 14.9 for SOC)	
**Ongoing clinical trials**								
**NCT01072981**	III	Adjuvant	722	Gemcitabine +/−Algenpantucel-L	28 × 1.8 Gy (+ 5-FU)	At 3rd dose	mOS	Recruitment completed
**NCT03767582**	I/rII	Neoadjuvant LA	30	Chemotherapy + Nivolumab + CCR2/5 dual antagonist +/− GVAX	5 × 6.6 Gy	2–4 weeks after chemotherapy and 2–3 weeks before ICIs	Safety/TILs response	Recruiting
**NCT04331041**	rII	Neoadjuvant LA	42	FFX or Gem-Np +/− Defactinib (FAK inhibitor)	5 × 10 Gy (MR-Linac)	Day 1, concurrent	PFS	Recruiting
**NCT04172532**	Ib/rII	Neoadjuvant LA	24	Pebosertib (DNA-PK inhibitor)	5 × ?Gy	Day 1	MTD, PFS	Recruiting
**NCT04106856** **(SHAPER)**	I	Neoadjuvant BR/LA	20	Losartan	15 × ?Gy	Day 14	Toxicity	Recruiting
**NCT02305186**	Ib/II	Neoadjuvant R/BR	68	Pembrolizumab	28 × 1.8Gy (+ capecitabine)	Concurrent	Number of TILs in resected tissue/Safety	Recruiting
**NCT05411094**	I	Unresectable LA	18	Olaparib (PARP inhibitor) + Durvalumab	NR	In cycle 2, concurrent	MTD	Recruiting
**NCT04247165** **(LAPTOP)**	Ib/II	BR/LA/M+	40	Gem-Np + Nivolumab + Ipilimumab	3 × 8 Gy (MR-Linac)	NR	Safety	Recruiting
**NCT03915678** **(AGADIR)**	II	M+ solid tumors (including PDAC)	247	Atezolizumab + BDB001 (TLR 7/8 agonist)	3–5 × 9–12 Gy	After 1st dose	ORR	Recruiting Basket trial
**NCT05116917** ** *(INFLUENCE)* **	II	M+	30	Nivolumab + Ipilimumab + Influenza vaccine	1 × 15 Gy	Day 1	ORR	Recruiting
**NCT05088889**	I	M+	10	Nivolumab + Ipilimumab +	3 × 8 Gy + 1 × 2 Gy for nonresponder	Day 1	ORR	Recruiting
**NCT04361162**	II	M+ (MSS)	30	Nivolumab + Ipilimumab	NR (3D)	Week 1 of 1st cycle	ORR	Active, not recruiting
**NCT04050085**	I	Refractory M+	6	Nivolumab + SD-101 (TLR-9 agonist)	5 × 6–10 Gy	Day 1	Safety	Active, not recruiting
**NCT03490760**	II	Refractory M+	39	Durvalumab	3 × 8 Gy	At week 5	PFS	Active, not recruiting
**NCT03161379**	II	Neoadjuvant BR	30	Cyclophosphamide + GVAX + Nivolumab	5 × 6.6 Gy	At 2nd dose	CD8 count (cell/ mm^3^) in the TME	Active, not recruiting
**NCT03563248** **(SU2C)**	rII	Neoadjuvant BR/LA	168	FFX +/− Losartan +/− Nivolumab	NR (SBRT)	Concurrent	R0 RR	Active, not recruiting
**NCT01595321**	I	Adjuvant	19	FFX + Cyclophosphamide + GVAX	5 × 6.6 Gy	After 1st dose	Toxicity	Active, not recruiting
**NCT04098432**	Ib/II	Neoadjuvant LA	15	FFX + Nivolumab	4 × 8 Gy	Before ICI	PFS	Active, not recruiting

RT = radiotherapy; CBR: clinical benefit rate; ORR: objective response rate; mOS: median overall survival; PFS: progression-free survival; DCR: disease response rate; GMCI: gene-mediated cytotoxic immunotherapy (aglatimagene besadenovec + valacyclovir); DMFS: distant metastasis-free survival; NS: not significant; TLR: toll-like receptor; MSS: microsatellite stable; LA: locally advanced; BR: borderline resectable; M+: metastatic; TME= tumor microenvironment; TIL = tumor-infiltrating lymphocytes; (R0) RR: (complete) resection rate; HDAC: histone deacetylase; FFX: FOLFIRINOX; Gem-Np: gemcitabine/Nab-paclitaxel.

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
