# Peer review of "Combination, Modulation and Interplay of Modern Radiotherapy with the Tumor Microenvironment and Targeted Therapies in Pancreatic Cancer: Which Candidates to Boost Radiotherapy?"

_cancers, 2023, doi:10.3390/cancers15030768_

Round 1

Reviewer 1 Report

The authors have written a well summarized review on therapy modalities using radiation and targeted approaches for cancer. I appreciate for their effort.

However they could also include couple of more topics such as using cancer targeting receptors such as carcinoembryonic antigen, CD44 receptors which as mostly expressed in various cancers including pancreatic. Also using new generation lipidic nanoparticles called ‘cubosomes’ for targeted delivery of radioactive drugs could also be a potential solution and area to be explored. Some reference articles such as https://doi.org/10.1021/acsami.1c21655; https://doi.org/10.1021/acs.molpharmaceut.2c00439 could be used for this literature.

Author Response

The authors have written a well summarized review on therapy modalities using radiation and targeted approaches for cancer. I appreciate for their effort.

Response: We would like to thank the reviewer for the positive and constructive comments that are very helpful to establish a more comprehensive and clearer manuscript.

  1. However they could also include couple of more topics such as using cancer targeting receptors such as carcinoembryonic antigen, CD44 receptors which as mostly expressed in various cancers including pancreatic. Also using new generation lipidic nanoparticles called ‘cubosomes’ for targeted delivery of radioactive drugs could also be a potential solution and area to be explored. Some reference articles such as https://doi.org/10.1021/acsami.1c21655; https://doi.org/10.1021/acs.molpharmaceut.2c00439 could be used for this literature.

Response:

The following paragraph has been added in the chapter « Combination of tRT », p.13 :

« Other methods are also under study to overcome the related-TME issues in PDAC by improving treatment delivery/efficacy, such as nanoparticles (NP). Recent successful clinical results were already obtained in PDAC for the following chemotherapy nanocarriers: nab-paclitaxel (Abraxane®) associated with gemcitabine and the nanoliposomal irinotecan (Onivyde®) associated with fluorouracil and folinic acid. NP-based delivery of targeted therapy is an interesting strategy which is under development for PDAC and other cancers. NP can also be used as local radiosensitizers, allowing to increase DNA damage and the production of reactive oxygen species (ROS). Several types of NP-radiosensitizers are studied, including in PDAC. A randomized phase I/ II clinical trial is currently recruiting in the USA for localized lung tumors and LA PDAC (n=100), aiming to compare the treatment efficacy of MR-guided SBRT +/- a gadolinium-based NP radiosensitizer (NCT04789486). Combinations of NP-based therapy with RT should be further investigated in the future. »

The following references have been added :

  • Caputo D, Pozzi D, Farolfi T et al. Nanotechnology and pancreatic cancer management: State of the art and further perspectives. World J Gastrointest Oncol. 2021 Apr 15;13(4):231-237. 
  • Jia, M., Zhang, D., Zhang, C. et al.Nanoparticle-based delivery systems modulate the tumor microenvironment in pancreatic cancer for enhanced therapy. J Nanobiotechnol 19, 384 (2021). 
  • Von Hoff DD, Ervin T, Arena FP et al. Increased survival in pancreatic cancer with nab-paclitaxel plus gemcitabine. N Engl J Med 2013; 369: 1691-1703.
  • Wang-Gillam A, Li CP, Bodoky G et al. NAPOLI-1 Study Group. Nanoliposomal irinotecan with fluorouracil and folinic acid in metastatic pancreatic cancer after previous gemcitabine-based therapy (NAPOLI-1): a global, randomised, open-label, phase 3 trial. Lancet. 2016 Feb 6;387(10018):545-557. 
  • Wang-Gillam A, Hubner RA, Siveke JT et al. NAPOLI-1 phase 3 study of liposomal irinotecan in metastatic pancreatic cancer: Final overall survival analysis and characteristics of long-term survivors. Eur J Cancer. 2019 Feb;108:78-87. 
  • Noubissi Nzeteu GA, Gibbs BF, Kotnik N et al. Nanoparticle-based immunotherapy of pancreatic cancer. Front Mol Biosci. 2022 Aug 29;9:948898.
  • Pramanik A, Xu Z, Ingram N et al. Hyaluronic-Acid-Tagged Cubosomes Deliver Cytotoxics Specifically to CD44-Positive Cancer Cells. Mol Pharm. 2022 Dec 5;19(12):4601-4611
  • Bilynsky C, Millot N, Papa AL. Radiation nanosensitizers in cancer therapy-From preclinical discoveries to the outcomes of early clinical trials. Bioeng Transl Med. 2021 Sep 23;7(1):e10256.
  • Chen Y, Yang J, Fu S, Wu J. Gold Nanoparticles as Radiosensitizers in Cancer Radiotherapy. Int J Nanomedicine. 2020 Nov 24;15:9407-9430. 
  • Wason M.S., Lu H., Yu L et al. Cerium oxide nanoparticles sensitize pancreatic cancer to radiation therapy through oxidative activation of the JNK apoptotic pathway. 2018;10:303.
  • Alhussan A, Palmerley N, Smazynski J et al. Potential of Gold Nanoparticle in Current Radiotherapy Using a Co-Culture Model of Cancer Cells and Cancer Associated Fibroblast Cells. Cancers (Basel). 2022 Jul 22;14(15):3586. 
  • Nano-SMART: Nanoparticles With MR Guided SBRT in Centrally Located Lung Tumors and Pancreatic Cancer (https://clinicaltrials.gov/ct2/show/NCT04789486 ) (accessed 18 January 2023)

Reviewer 2 Report

1.       What knowledge gap are the authors covering in this review? How the current review stands different from the available literature?

2.       Section 3 needs to be reformatted or rearranged. Multiple subsections in the name “Effects of RT”??

3.       Please include a section about the limitations of RT and its long-term side effects.

4.       Figure 1 needs to improve the visibility of the text.

5.       There are a lot of opportunities to include illustrations in this review. The authors suggested including some more illustrations that will make this review more interactive.

Author Response

We would like to thank the reviewer for the positive and constructive comments that are very helpful to establish a more comprehensive and clearer manuscript.

  1. What knowledge gap are the authors covering in this review? How the current review stands different from the available literature?

Response:

To our knowledge, we are the first to provide a detailed review of the known RT-induced effects/modulation on the different components of the TME of PDAC. Furthermore, this review provides an up-to-date summary and emphasizes the perspective of targeted therapy combination with RT in PDAC in order to improve and increase further research in this field.

  1. Section 3 needs to be reformatted or rearranged. Multiple subsections in the name “Effects of RT”??

Response:

The layout of the chapter has been rearranged in order to improve clarity.

  1. Please include a section about the limitations of RT and its long-term side effects.

Response:

The following paragraphs have been added in the chapter 2 (p. 3) :

« The role of RT in PDAC has been intensely debated over the past 30 years. Although RT is a treatment option validated by international guidelines, the exact role of RT in PDAC remains to be further explored and validated in randomized clinical trials. »

«  If the use of conventional chemoradiotherapy (CRT) is currently declining following the disappointing results of the available randomized phase III trials (17,18), altered dose prescriptions and new RT techniques are now showing promising oncological results associated with tolerable acute and mid-term toxicity, although long-term results remain to be studied. »

The following references have been added :

  • Tempero MA, Malafa MP, Al-Hawary M et al. Pancreatic Adenocarcinoma, Version 2.2021, NCCN Clinical Practice Guidelines in Oncology. J Natl Compr Canc Netw 2021; 19: 439-457.
  • Khorana AA, McKemin SE, Berlin J, et al. Potentially curable pancreatic adenocarcinoma: ASCO clinical practice guideline update. J Clin Oncol. 201937: 23,2082-2088.

  1. Figure 1 needs to improve the visibility of the text.

Response: Figure 1 has been enlarged and the text size increased (p16).

  1. There are a lot of opportunities to include illustrations in this review. The authors suggested including some more illustrations that will make this review more interactive.

Response: A Figure 2 has been created in order to illustrate the selected promising clinical perspectives of tRT in PDAC.